# Ramadan Observance Is Associated with Impaired Kung-Fu-Specific Decision-Making Skills

**DOI:** 10.3390/ijerph18147340

**Published:** 2021-07-09

**Authors:** Anis Saddoud, Aïmen Khacharem, Cyrine H’Mida, Khaled Trabelsi, Omar Boukhris, Achraf Ammar, Cain C. T. Clark, Jordan M. Glenn, Hamdi Chtourou, Mohamed Jarraya, Thomas Rosemann, Beat Knechtle

**Affiliations:** 1High Institute of Sport and Physical Education of Sfax, University of Sfax, Sfax 3000, Tunisia; anissaddoud75@gmail.com (A.S.); sirinehmida@hotmail.fr (C.H.); trabelsikhaled@gmail.com (K.T.); omarboukhris24@yahoo.com (O.B.); h_chtourou@yahoo.fr (H.C.); jarrayam@yahoo.fr (M.J.); 2Research Laboratory: Education, Motricity, Sport and Health, EM2S, LR19JS01, University of Sfax, Sfax 3000, Tunisia; 3UFR SESS-STAPS, Paris-East Créteil University, LIRTES (EA 7313), 94000 Créteil, France; aimen.khacharem@gmail.com; 4DeVisu (EA 2445), Polytechnic University of Hauts-de-France, 59313 Valenciennes, France; 5Physical Activity, Sport, and Health, UR18JS01, National Observatory of Sport, Tunis 1003, Tunisia; 6Institute of Sport Science, Otto-von-Guericke University, 39106 Magdeburg, Germany; ammar.achraf@ymail.com; 7Interdisciplinary Laboratory in Neurosciences, Physiology and Psychology: Physical Activity, Health and Learning (LINP2), UFR STAPS, UPL, Paris Nanterre University, 92000 Nanterre, France; 8Centre for Intelligent Healthcare, Coventry University, Coventry CV1 5FB, UK; ad0183@coventry.ac.uk; 9Department of Health, Exercise Science Research Center, Human Performance and Recreation, University of Arkansas, Fayetteville, AR 72701, USA; jordan.mckenzie.glenn@gmail.com; 10Neurotrack Technologies, 399 Bradford St, Redwood City, CA 94063, USA; 11Institute of Primary Care, University of Zurich, 8006 Zurich, Switzerland; thomas.rosemann@usz.ch; 12Medbase St. Gallen Am Vadianplatz, 9000 St. Gallen, Switzerland

**Keywords:** intermittent fasting, decision making, sleep, sleepiness, fatigue, athletes

## Abstract

The aim of the present study is to evaluate the effect of Ramadan observance (RAM) on decision-making in Kung-Fu athletes. Fourteen male Kung-Fu athletes (mean age = 19 ± 3 years) completed two test sessions: before Ramadan (BR) and at the end of Ramadan (ER). In the afternoon of each session (between 16:00 h and 18:00 h), participants completed: Epworth Sleepiness Scale (ESS), Profile of Mood States (POMS), and Pittsburg Sleep Quality Index (PSQI). Subjects also reported subjective fatigue, alertness, and concentration. Additionally, all participants performed video-based decision-making tasks (i.e., reaction time and decision-making). Results indicated that reaction time decreased by 30% at ER vs. BR (*p* < 0.01). However, decision-making decreased by 9.5% at ER vs. BR (*p* < 0.05). PSQI results indicated sleep quality score, sleep duration, and sleep efficiency were negatively affected at ER compared to BR (*p* < 0.05). ESS was higher at ER compared to BR (*p* < 0.05). In addition, fatigue scores, estimated by the POMS and current subjective feelings (i.e., fatigue, concentration, and alertness), were also negatively affected at ER compared to BR (*p* < 0.05). In conclusion, Ramadan observance was associated with an adverse effect on sleep and decision making, as well as feelings of fatigue, alertness, and concentration.

## 1. Introduction

Every year, Muslims abstain from eating, drinking, and sexual intercourse, from sunrise to sunset, over a period that lasts 29 or 30 days, i.e., the month of Ramadan [1]. Ramadan observance (RAM) has been reported to yield important negative changes in physical performance [2,3], body composition [4], and eating habits [5,6]. Moreover, the continuance of training during RAM (often scheduled at night), in association with nocturnal mealtimes [7], could also affect negatively sleep-wake patterns [8,9]. Additionally, some studies have shown increased fatigue estimated by the Profile of Mood State (POMS) scale (without any changes in tension, depression, anger, confusion, and vigor estimated by the POMS) [10,11,12], by the Hooper questionnaire [13,14] and by the rating of perceived exertion [15].

However, despite the evidence in the extant literature demonstrating that cognitive performance may be impaired by the typical changes related to meal timing and macronutrient composition [16,17], dehydration [18,19], and sleep loss [20,21], a dearth of attention has been dedicated to understanding the effect of RAM on cognitive function in athletes [9,22]. For example, Tian et al. [22] evaluated the effect of RAM on cognitive function using computerized neuropsychological testing, according to time of day in healthy Muslim athletes, and reported that in the morning (09:00 h), detection and identification performances were better during RAM vs. after RAM while in the afternoon (16:00 h), verbal learning and short-term memory performances were better after RAM compared to during RAM [22]. These findings indicate that the effects of RAM on cognitive function were heterogeneous and domain-specific with respect to the time of day [22]. Conversely, Chamari et al. [9] showed that RAM did not affect cognitive performance in trained cyclists from the Middle East.

To date, to the authors’ knowledge, no published research has examined the effects of RAM on sport-specific decision-making, which is a vital aspect of perceptual skill. Moreover, in combat sports, the ability to perceive and process opponent intention is important for evasion, to initiate an effective counter-attack [23]. Indeed, in such sports, the time for making decisions is a condition that should always be considered, given that actions often take place within highly complex and time-constrained situations [24].

Therefore, the aim of this study was to evaluate the effect of RAM on decision-making tasks, as well as sleep, sleepiness, mood states, and feelings of fatigue in combat sports. In the present study, Kung-Fu was chosen because of its dynamic nature, where adaptation and decision making are characteristics [23]. Given that RAM is generally accompanied by a myriad of changes (e.g., meal timing, dehydration, sleep loss, mood, reaction time, and cognitive performance), we hypothesized that the decision-making performance (decision time and decision accuracy) of Kung-Fu athletes would be negatively affected during RAM compared to the pre-RAM period. Additionally, we hypothesized that RAM would be associated with a decline in sleep quality and the mood state of Kung-Fu athletes.

## 2. Materials and Methods

### 2.1. Participants

Fourteen male Kung-Fu athletes (mean age = 19 ± 3 years) volunteered for this study with experience ranging between 3 and 6 years. Eight of them had a black belt and the remaining had a brown belt. They participated in regional and national high-level competitions.

After receiving a description of the protocol, potential risks, and benefits of the study, participants gave their written consent to participate in this investigation. The study was conducted according to the declaration of Helsinki and the protocol was fully approved by the Research Ethics Committee before the commencement of the assessments. The study was carried out in Tunisia, when Ramadan started on the 5 May and concluded on the 3 June 2019. The length of each fasting day during Ramadan observance was approximately ~16 h. The criteria for participant inclusion in this study were as follows: all participants (I) continued training during Ramadan, (II) were non-smokers, (III) did not have pathological sleep disorders, and (VI) did not consume alcohol.

### 2.2. Material and Apparatus

For the video stimuli, a Kung-Fu fight that opposed of two male athletes (red and blue) was recorded by a digital video camera (Sony DCR_TRV10). The recording took place in a judo-exercise gymnasium, where the video camera was positioned at the height of 1.7 m and at a distance of 5 m from the athletes’ initial position. From the recording of the entire fight, 35 scenarios were chosen (with the collaboration of two Kung-Fu expert coaches—mean age = 45 yrs) as the stimuli to be presented in the experimental condition. During the experiment, each fight scenario lasted between 4 and 10 s and was then automatically occluded by a black frame at the moment two other expert coaches (mean age = 44 years) considered it appropriate for a decision to be made. Participants were asked to verbally indicate, as quickly as possible, “What offensive action should the athlete with the RED uniform do at that moment?”. The options were: (a) upper kick, (b) middle kick, or (c) lower kick.

The experimental test was performed indoors, without external interference. All the scenarios were displayed on a white wall to create a large 4 m × 3 m projected image using a Sanyo PDG-DET100L Projector (Sanyo 231 Electric CoLtd., Osaka, Japan). The participants were positioned three meters from the screen and seated with a clear view of the screen. Players were introduced to five practical scenarios prior to the test for familiarization.

### 2.3. Experimental Design

A familiarization session was performed before Ramadan. Participants conducted two test sessions: before two weeks of Ramadan (BR) and at the end of the last week of Ramadan (i.e., the 28th day of Ramadan) (ER). In the afternoon of each session (between 16:00 h and 18:00 h), participants completed the Epworth Sleepiness Scale (ESS) [25], followed by the POMS [26] and the Pittsburg Sleep Quality Index (PSQI) [27] questionnaires. In addition, participants rated their current subjective feelings for their levels of ‘fatigue’, ‘alertness’, and ‘concentration’ on a 100 mm visual analogue scale [28]. Finally, the participants performed a decision test (Figure 1).

### 2.4. Procedure

Each participant was invited to follow the video sequences presented via a video projector. Five sequences were utilized during the familiarization session. The remaining sequences (*n* = 30) were used to evaluate the athletes’ decision making.

After the onset of an offensive action, and simultaneously when the video was stopped, participants were asked to determine if the action was orientated on the upper, the middle, or the lower body part of the athlete. Each participant was asked to decide as quickly and as accurately as possible the best response by clicking on the keyboard (three keys for the three possible solutions) of the computer. The decision time and accuracy were automatically recorded using C# programming language.

### 2.5. The Profile of Mood States (POMS)

Subjective mood states were assessed using the French version of the POMS questionnaire [26]. This is a self-report questionnaire consisting of 65 adjectives designed to evaluate six states (i.e., tension, depression, anger, vigor, fatigue, and confusion). Responses to each item range from 0 to 4, with higher scores indicating a more negative mood (0 indicates “Not at all” and 4 indicates “Extremely”). The total score of the POMS was calculated as follows: total score = (tension + depression + anger + fatigue + confusion)—vigor.

### 2.6. The Epworth Sleepiness Scale (ESS)

This ESS questionnaire was used to assess subjective sleepiness [25]. The participant responded how likely they are to doze in eight different daily situations on a 4-point scale. ESS scores were interpreted based on the following references [29]: 0–5 indicates lower than normal daytime sleepiness, 6–10 indicates higher than normal daytime sleepiness, 11–12 indicates mild excessive daytime sleepiness (EDS), 13–15 indicates moderate EDS, and 16–24 indicates severe EDS.

### 2.7. The Pittsburg Sleep Quality Index (PSQI)

The sleep quality was assessed by the PSQI [30], which has been extensively validated in different cultures and populations [31]. The Arabic-validated version of the PSQI was utilized in the present study [27]. The PSQI questionnaire was used to assess subjective sleep quality over the previous month (i.e., one-month BR and during the month of Ramadan). The questionnaire was composed of 19 questions, each representing one of the seven components of sleep quality: subjective sleep quality, sleep latency, sleep duration, sleep efficiency, sleep disturbance, sleep medication intake, and daytime dysfunction. Each component score was rated on a 3-point scale, leading to a sum of up to 21 points. PSQI scores >5 and ≤5 indicated, respectively, poor and good sleep qualities [30].

### 2.8. Perceptual Measures

Participants rated their subjective feelings before the test session to estimate the level of attention, concentration, and fatigue using a 100 mm visual analogue scale. The participant marked the point that signified his perception of his current state for each question. The visual analogue scale score was determined by measuring the distance in millimeters (mm), from the left side of the end of the line to the player’s marked line. In addition, the sleep duration of the night preceding the test session, during the two periods of the study (i.e., BR and ER), was estimated by each participant. Furthermore, participants were also asked about their subjective sleep quality for each night preceding the test session using a scale ranging from “1” (very poor quality) to “5” (very good quality).

### 2.9. Statistical Analysis

All statistical tests were processed using STATISTICA Software (StatSoft, Paris, France). Mean and SD (standard deviation) values were calculated for each variable.

The software G*Power [32] was used, as well as a priori to calculate the necessary minimum sample size, based on procedures suggested by Beck [33]. Values for α were set at 0.05 and for power at 0.8. Based on an earlier study by Boukhris et al. [15] and discussions between the authors, the likely effect sizes were estimated as 0.45. In total, to reach the desired power, for a pre- vs. post-repeated measures study design, data from at least twelve participants were deemed to be sufficient to minimize the risk of incurring a type 2 statistical error.

The Shapiro-Wilk test revealed that sleep latency, total score of PSQI, mood states (i.e., tension, anger, vigor, fatigue, and confusion), ESS, decision test, fatigue, and concentration were normally distributed and were analyzed using a paired sample t-test in order to compare them between BR and ER. However, when the Shapiro-Wilk test was significant (*p* < 0.05), pairwise comparisons were conducted using a Wilcoxon test. The statistical calculation was performed for a two-tailed test. Effect size was calculated using Cohen’s d for all parameters. Cohen’s d of 0.2, 0.5, and 0.8 represent small, moderate, and large effect sizes, respectively [34]. Statistical significance was accepted, for all analyses, at the level of *p* < 0.05.

## 3. Results

### 3.1. Profile of Mood States (POMS)

The results of the POMS are illustrated in Table 1. There was no significant difference between BR and ER for anxiety, confusion, depression, tension, vigor, and total score. However, statistical analysis showed that fatigue increased at ER vs. BR.

### 3.2. Epworth Sleepiness Scale (ESS)

Statistical analysis showed that ESS increased significantly from 5.6 ± 2.9 at BR to 7.1 ± 3.5 at ER (21%, *p* = 0.004, and *d* = 0.46).

### 3.3. The Pittsburgh Sleep Quality Index (PSQI)

The results of the PSQI questionnaire are presented in Table 2. Sleep quality scores were higher during RAM compared to BR. However, sleep duration and sleep efficiency were lower during RAM compared to BR.

There was no significant difference between BR and during RAM for sleep latency (*p* = 0.14, *d* = 0.45), sleep disturbance, daytime dysfunction, the use of sleeping medications, and total PSQI.

### 3.4. Perceptual Measures

The obtained results of the subjective responses are provided in Table 3. Compared to BR, sleep duration, sleep quality, attention, and concentration decreased significantly at ER. However, fatigue increased significantly at ER compared to BR.

### 3.5. Decision-Making: Response Accuracy and Reaction Time

As illustrated in Figure 2, statistical analysis showed that the response accuracy significantly decreased from BR to ER (20.0 ± 2.4 vs. 18.1 ± 3.0; *p* = 0.02, *d* = 0.69, and 95% IC = 0.21–3.50).

Statistical analysis showed that reaction time significantly decreased at ER compared to BR (1.0 ± 0.4 vs. 0.7 ± 0.3; *p* = 0.002, *d* = 0.84, and 95% IC = 0.09–0.33) (Figure 3).

## 4. Discussion

The purpose of the present study was to evaluate the difference between BR and ER on decision making (i.e., decision accuracy and time) as well as on sleep, sleepiness, mood states, and feelings of fatigue in Kung-Fu athletes. The present results showed that athletes’ decision-making (i.e., response accuracy and reaction time) BR was better than at the ER. Similarly, Cherif et al. [35] reported that daylight fasting could negatively affect many aspects of physical performance and mental health, including coping and decision-making strategies. Thus, the cognitive and physical aspects of RAM can represent a notable challenge for Muslim athletes during Ramadan [35]. In this context, the literature has demonstrated that alertness, memory, reaction time, and psychomotor performance may be deleteriously affected by RAM [14,36,37,38,39]. However, the present findings were not in agreement with previous data, showing that cognitive function was unaffected by RAM [9,40,41].

Sleep disruption is among the many factors that could negatively affect the decision-making of athletes during RAM. Indeed, Venkatraman et al. [42] suggested that decision-making was altered by sleep loss, which modulates activation in the nucleus accumbens and insula brain regions, related to risky decision making and emotional processing. In accordance with previous studies [1,8,13,43,44], sleep duration during the night before the test was less during vs. BR. Similarly, the average sleep duration at BR (reported by the PSQI) was higher than during RAM. This sleep loss may be attributed to the shift of training and meal timing during the night of RAM, which could lead to 1–2 h of sleep loss per day [45,46]. Additionally, this sleep loss could be the cause of an increase of daytime sleepiness [8,46]. Indeed, the current study demonstrated that daytime sleepiness was increased at ER vs. BR. However, this increase of sleepiness level indicates higher normal daytime sleepiness; thus, sleepiness was likely not a factor that contributed to the alteration of decision-making during RAM [47]. Indeed, the alteration of decision-making during RAM could be attributed to sleep loss [21]. In support of this claim, it has been reported that sleep deprivation leads to a slowing of reaction times and attentional deficits, particularly in the late afternoon [14]. Additionally, it has been reported that an impairment of sleep could affect the fatigue and supercompensation response in well-trained athletes [21]. Furthermore, sleeping less than the recommended duration (i.e., 9–10 h/day [48] or 8 h/day [49]) could provoke deficits in cognitive function [14]. In fact, the participants in the present study slept ~6.3 h during RAM, which is less than the recommended duration of sleep. However, other studies reported that when sleep duration was unchanged during RAM, cognitive performance was unaffected [9,41]. When the sleep disruptions persist throughout RAM, mood swings and a higher feeling of fatigue are often reported [50], thus negatively affecting afternoon performance in many sports activities. In the present study, RAM was associated with negative changes in fatigue (estimated by the POMS questionnaire), attention, and concentration. Concordant with the present results, some previous studies have reported increased subjective ratings of fatigue via the POMS [11,12] and the Hooper questionnaire [13,14]. Therefore, the alteration of decision-making apparent during RAM could be attributed to the decreased attention and concentration, as well as increased feelings of fatigue, possibly due to sleep disruptions. However, in the present study, the decrease of the decision time at the ER in comparison with BR was not accompanied with an improvement in decision making.

In accord with the aim of this study, to evaluate the effect of RAM on decision-making in Kung-Fu athletes, we have presented a novel addition to the literature that helps to elucidate the impact of Ramadan on combat sports athletes. Indeed, this represents the principal strength of the study and provides a stepping stone for researchers and practitioners to further examine the RAM phenomenon. Despite the novelty of our study, there are some limitations that warrant consideration. Indeed, the principal limitation is that the present study is observational, and thus, no causal inferences can be made. Next, the lack of objective sleep measurement may have resulted in some underestimation in our study. Moreover, despite the advantages of using video for exploring decision-making and perceptual expertise in sports, this tool suffers from its inability to faithfully reproduce the environment of athletes in performance situations [51]. In addition, athletes often react to the video from the moment a visual stimulus is presented, while a natural environment offers multiple sources of information including visual, auditory, or kinesthetic [52]. As the participants of the present study were Arabic native speakers, and as to the best of our knowledge no previous study has validated the POMS in the Arabic language, another limitation for the current study was the utilization of the French version of the POMS. However, all participants were students and the French language is taught from the third primary year to university level. Therefore, we are confident that all participants satisfactorily understood the French version of the POMS questionnaire. The results of the present study could not be generalized to all populations. In fact, our data and conclusions only refer to the male population, thus highlighting the need for further work specific to females. Additionally, the time of day could influence performance, attention, and fatigue according to an athlete’s circadian preferences [53,54,55]. Furthermore, there is no control for scheduled times of sleeping habits, eating, meal composition, training load, and training. This is difficult to altogether achieve during Ramadan, but future studies should at least control the training load during Ramadan. Finally, the present study was not powered to discern the categorized differences between athletes, BR vs. ER. We therefore advocate that future, suitably powered, trials seek to discern differences across the categories of sleep, sleepiness, mood states, feelings of fatigue, and Kung-Fu specific decision-making skills.

## 5. Conclusions

The present study demonstrated that RAM was associated with reductions in decision making. The alteration of decision-making during RAM could be attributed to a reduced attention and concentration, as well as increased feelings of fatigue, which in turn were likely influenced by sleep disruptions. Further work, particularly experimental study designs, are warranted to more firmly elucidate the impact of fasting on sleep, sleepiness, mood states, feelings of fatigue, and Kung-Fu specific decision-making skills.

## Figures and Tables

**Figure 1 ijerph-18-07340-f001:**
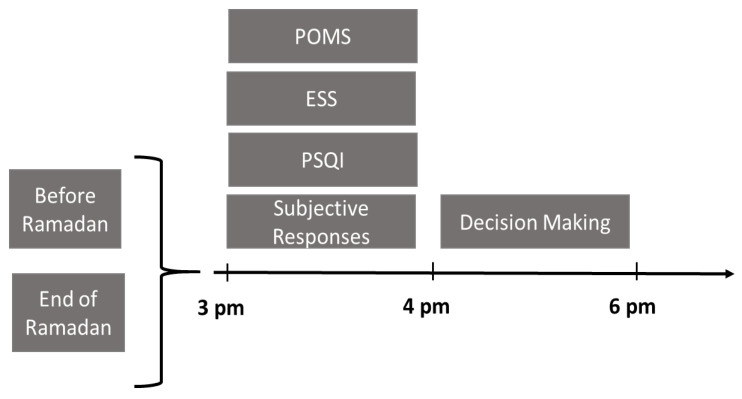
The schematic representation of the experimental design.

**Figure 2 ijerph-18-07340-f002:**
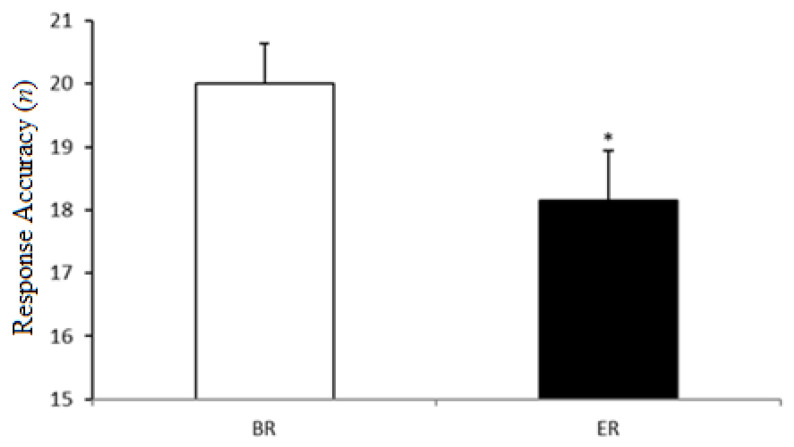
The response accuracy recorded before Ramadan (BR) and at the end of Ramadan (ER). *: Significant difference compared to BR.

**Figure 3 ijerph-18-07340-f003:**
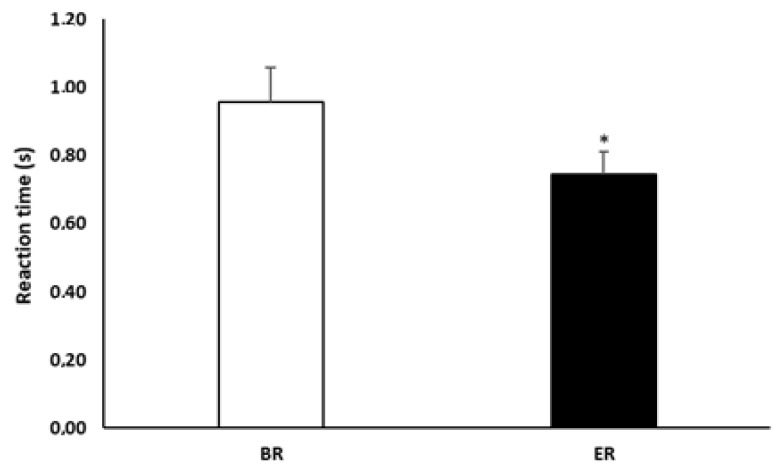
Reaction time recorded before Ramadan (BR) and at the end of Ramadan (ER). *: Significant difference compared to BR.

**Table 1 ijerph-18-07340-t001:** The results of the Profile of Mood States recorded before (BR) and at the end of Ramadan (ER).

Variables	BR	ER	Test	*p-*Value	Cohen’s *d*	95% IC
Tension (A.U)	9 ± 4	8 ± 4	T = 0.80	0.4	0.25 (moderate effect)	−1.31–2.88
Depression (A.U)	7 ± 7	6 ± 5	Z = 0.44	0.6	0.16 (small effect)	−2.48–4.48
Anger (A.U)	9 ± 6	10 ± 5	T = −0.56	0.5	0.18 (small effect)	−4.10–2.39
Vigor (A.U)	19 ± 4	18 ± 6	T = 1.07	0.3	0.19 (small effect)	−1.43–4.29
Fatigue (A.U)	5 ± 4	8 ± 5 *	T = −2.52	0.02	0.66 (large effect)	−5.56–−0.43
Confusion (A.U)	7 ± 4	7 ± 4	T = 0.42	0.6	0.04 (small effect)	−0.87–1.30
Total score (A.U)	18 ± 24	23 ± 21	T = −1.17	0.2	0.22 (moderate effect)	−14.37–4.22

*: significant difference compared to BR; A.U: arbitrary unit.

**Table 2 ijerph-18-07340-t002:** Sleep parameters recorded by the PSQI questionnaire before Ramadan (BR) and at the end of Ramadan (ER).

Variables	BR	ER	Test	*p*-Value	Cohen’s *d*	95% IC
Sleep quality (A.U)	1.1 ± 0.9	1.9 ± 0.9 *	Z = 2.34	0.01	0.88 (large effect)	−1.19–−0.23
Sleep latency (min)	14.6 ± 9.0	19.1 ± 10.6	T = −1.56	0.14	0.45 (moderate effect)	−10.70–1.70
Sleep duration (h)	7.8 ± 1.1	6.3 ± 1.9 *	Z = 2.41	0.01	1.26 (large effect)	0.42–2.64
Sleep efficiency (%)	97.1 ± 5.0	87.2 ± 13.0 *	Z = 2.08	0.03	1.01 (large effect)	1.25–18.60
Sleep disturbances (A.U)	1.2 ± 0.4	1.1 ± 0.9 *	Z = 0.36	0.7	0.14 (small effect)	−0.35–0.49
Daytime dysfunction (A.U)	1.1 ± 1.3	0.9 ± 0.9	Z = 1.34	0.1	0.17 (small effect)	−0.13–0.70
The use of sleeping medications (A.U)	0 ± 0	0 ± 0	–	–	–	–
Total score of PSQI (A.U)	5.2 ± 2.4	6.4 ± 2.9	T = −1.70	0.1	0.45 (moderate effect)	−2.75–0.32

*: Significant difference in comparison with BR; A.U: arbitrary unit.

**Table 3 ijerph-18-07340-t003:** The perceived levels of subjective attention, fatigue, sleep duration, sleep quality, and concentration recorded before Ramadan (BR) and at the end of Ramadan (ER).

Variables	BR	ER	Test	*p-*Value	Cohen’s *d*	95% IC
Sleep duration (h)	7.3 ± 1.2	6.1 ± 1.8 *	Z = 2.13	0.03	0.78 (large effect)	0.004–0.09
Sleep quality (A.U)	3.7 ± 0.9	2.6 ± 1.2 *	Z = 2.40	0.01	1.03 (large effect)	0.30–1.83
Attention (A.U)	73.9 ± 20.8	47.5 ± 19.3 *	Z = 2.62	0.008	1.31 (large effect)	8.40–44.45
Concentration (A.U)	75.4 ± 16.7	53.9 ± 18.6 *	T = 2.84	0.01	1.21 (large effect)	5.15–37.69
Fatigue (A.U)	38.6 ± 25.3	60.4 ± 24.9 *	T = −2.58	0.02	0.86 (large effect)	−39.99–−3.57

*: Significant difference in comparison with BR; A.U: arbitrary unit.

## Data Availability

The data that support the findings of this study are available on request from the first author, Anis Saddoud.

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
