# Peer review of "Ramadan Observance Is Associated with Impaired Kung-Fu-Specific Decision-Making Skills"

_ijerph, 2021, doi:10.3390/ijerph18147340_

Round 1
Reviewer 1 Report
GENERAL COMMENTS
The study aims to assess accuracy and reaction time in a sample of 14 Kung-fu males during Ramadan. The article is interesting, well-written and seems to investigate an uncovered topic.
However, there are many inaccuracies in the questionnaires explanations and utilizations and the results explanations are not matching the run statistical analysis. Indeed, many time the authors affirm to have discovered the effects of Ramadan fasting. Instead, they simply find a difference between pre- and post-Ramadan period.
If the author would discover an effect of Ramadan fasting, they should run more and different analysis.
In addition to improving the material and results section and running further analysis, the authors should review the discussion part considering the correct wording of their results (differences and not effects).
MATERIALS AND METHODS
- Line 90: in addition to the practised year, is it possible to know the level of the subjects and if they participate at the same competition level? Maybe the level could influence the results.
- Line 93: maybe the approbation number is needed.
- Why did the authors opt for two questionnaires investigating sleepness? Results of the Karolinska questionnaire are not reported.
- Are the questionnaires also available in the authors' language? If so, the authors should also insert the reference for the translation studies. N the contrary, they should specify how did they administrate the questionnaires.
- The reference period is missing for the questionnaires descriptions.
- It is not clear when ER corresponds to. Did the tests and questionnaires take place immediately after Ramadan, after a week, a month, etc.?
- Line 156: what does EDS refer to?
- A description of the analogue scale is not provided.
- POMS: it is not clear how the total value is obtained.
- Lines 173-178: im opinion, it is not necessary to repeat twice the normal distributed variables. The authors can simply say that the normally distributed values were analyzed with the t-test.
- In the statistical analysis is not written that the authors compared BR and ER.
- CIs are not specified.
- The authors just compared the obtained results before and after Ramadan. Did they evaluate the possibility of running correlation analysis or evaluating the results based on categorical variables (such as good and bad sleepers or ESS categorization)?
- Furthermore, did the authors consider adjusting the analysis for sleep, fatigue, sleepiness, etc.?
RESULTS
- In my opinion, authors should respect the order in Figure 1 to present the results.
- Lines 183 197: the authors did not evaluate the effects, but the differences between before and after Ramadan.
- Moreover, they did not investigate fasting (such as hours since last meal) or use the fasting condition as a covariate, and it could not maybe be correct saying "effects of Ramadan fasting."
- The punctuation should be more consistent: sometimes there is the space between = and sometimes not. Furthermore, P and D should be in italics.
- Tables are not recalled in the text.
- Line 194: why the authors wrote during and not BR; maybe they should be more precise in defining the evaluated periods.
- Table 2: the authors partially used PSQI components and partly the free-answers of the questionnaire. Maybe, it may be coherent to report all the seven components in addiction to sleep latency, duration and efficiency.
- 4. Subjective responses to the questionnaires:
- The title is equivocal because all the answer are subjective;
- Sleep duration and quality are part of the PSQI questionnaire;
- KSS results are reported without being explained in the methods section.
- Are these data referring to the visual analogue scale?
- 5 Decision-making (response accuracy): in this paragraph, also reaction time analysis is included; thus, the title is not inclusive of all the reported data.
- Figure 2:
- it is not recalled in the text;
- the measuring unit is missing;
- bars should be of different colours.
DISCUSSION
- Line 224, Another time, the authors evaluated the differences between ER and BR rater than the effects.
- Line 240, why did the authors used during and not ER?
- Lines 245-248, did the authors write this sentence referring to their data or those of another study. In the case of the first hypothesis, this could not be said based only on the showed statistical analysis.
- Line 264, the authors did not evaluate the association in the statistical analysis.
- Line 271, authors did not conduct an intervention study; thus, they did not need a control group excluding the possibility of a not-Muslim or not-observing Ramadan control group.
- The authors should consider that their analysis does not involve women, and their data and conclusion only refer to men population.
- Also, time of day could influence performance, attention and fatigue. I suggest the authors also consider the following papers: Mulè et al., 2020, doi: 10.1007/s11332-019-00610-9; Roveda et al., 2020 doi: 1080/07420528.2020.1729787; Montaruli et al., 2019, doi: 10.1080/07420528.2019.1652831.
Author Response
Reviewer 1:
GENERAL COMMENTS
The study aims to assess accuracy and reaction time in a sample of 14 Kung-fu males during Ramadan. The article is interesting, well-written and seems to investigate an uncovered topic.
However, there are many inaccuracies in the questionnaires explanations and utilizations and the results explanations are not matching the run statistical analysis. Indeed, many time the authors affirm to have discovered the effects of Ramadan fasting. Instead, they simply find a difference between pre- and post-Ramadan period.
If the author would discover an effect of Ramadan fasting, they should run more and different analysis.
In addition to improving the material and results section and running further analysis, the authors should review the discussion part considering the correct wording of their results (differences and not effects).
MATERIALS AND METHODS
- Line 90: in addition to the practised year, is it possible to know the level of the subjects and if they participate at the same competition level? Maybe the level could influence the results.
Answer: All the athletes have the same level, we carry this study on a homogeneous population. Therefore, there isn’t any influence on results.
The following sentences are added in section 2.1 on page 2: “Eight of them had black belt and the remaining had brown belt. They participated in regional and national high-level competitions.”
- Line 93: maybe the approbation number is needed.
Answer: We have added the approval number of the Ethical committee:
“The study was conducted according to the declaration of Helsinki and the protocol was fully approved by the Research Ethics Committee (CPP: 0102/2020), before the commencement of the assessments.”
Please see changes made in the revised version.
- Why did the authors opt for two questionnaires investigating sleepness? Results of the Karolinska questionnaire are not reported.
Answer: The reviewer is right, the Epworth sleepiness scales (ESS) and Karolinska sleepiness scale (KSS) are used for the same purpose (the sleepiness assessment). For this reason, KSS questionnaire was removed. The corresponding changes are illustrated in the manuscript.
- Are the questionnaires also available in the authors' language? If so, the authors should also insert the reference for the translation studies. N the contrary, they should specify how did they administrate the questionnaires.
Answer: Yes the questionnaire is available in the authors’ language. We have now added the following sentence:
"The Arabic validated version of the PSQI was utilized in the present study (Suleiman et al. 2010)."
Please see changes made in the Materials and Methods section.
To the best of our knowledge, no previous study has validated the profile of mood states (POMS) in Arabic and we utilized the version validated by Cayrou et al. (2003) in the French language. French is a second language in the authors’ country and most peoples have a good knowledge of French understanding and speaking.
Otherwise, we added this information in the discussion section as a limitation for the study.
The following sentence was added:
“As the participants of the present study were Arabic native speakers and as to the best of our knowledge no previous study has validated the POMS in Arabic language, another limitation for the current study was the utilization of the French version of the POMS. However, all participants were students and French language is taught from the third primary year to university level. Therefore, we are confident that all participants satisfactorily understood the French version of the POMS questionnaire."
Please see changes made in the discussion section.
- The reference period is missing for the questionnaires descriptions.
Answer: Corrected. Please see changes made in the text.
- It is not clear when ER corresponds to. Did the tests and questionnaires take place immediately after Ramadan, after a week, a month, etc.?
Answer: This part was modified as follow:
“Participants conducted two test sessions: before Ramadan (BR) and at the end of the last week of Ramadan (i.e., 28th day of Ramadan) (ER).”
Please see changes made in the revised version.
- Line 156: what does EDS refer to?
Answer: EDS refers to Excessive Daytime Sleepiness.
We added this information in the revised version.
- A description of the analogue scale is not provided.
Answer: We have now added a description of the analogue scale.
The following sentences were added:
“2.8. Perceptual measures
Participants rated their subjective feeling before the test session to estimate the level of attention, concentration and fatigue using a 100 mm visual analogue scale. The participant marked the point that signified his perception of his current state for each question. The visual analogue scale score was determined by measuring the distance in millimetre (mm), from the left-side of the end of the line to the player’s marked line. In addition to that, the sleep duration of the night preceding the test session during the two periods of the study (i.e., BR and ER) was estimated by each participant. Furthermore, participants were also asked about their subjective sleep quality for each night preceding the test session using a scale ranging from “1” (very poor quality) to “5” (very good quality).”
Please see changes made in the revised version.
- POMS: it is not clear how the total value is obtained.
Answer: This information was added:
“The total score of the POMS was calculated as follows: Total score = (tension + depression + anger + fatigue + confusion) - vigor”
Please see changes made in the revised version.
- Lines 173-178: im opinion, it is not necessary to repeat twice the normal distributed variables. The authors can simply say that the normally distributed values were analyzed with the t-test.
Answer: Corrections made as suggested. Please see changes made in the revised version.
- In the statistical analysis is not written that the authors compared BR and ER.
Answer: This information was added in the statistical analysis. Please see changes made in the revised version.
- CIs are not specified.
Answer: The CIs were added. Please see changes made in the revised version.
- The authors just compared the obtained results before and after Ramadan. Did they evaluate the possibility of running correlation analysis or evaluating the results based on categorical variables (such as good and bad sleepers or ESS categorization)? Furthermore, did the authors consider adjusting the analysis for sleep, fatigue, sleepiness, etc.?
Answer: We verify the correlation analysis. However, there was no significant relationship between all parameters.
RESULTS
- In my opinion, authors should respect the order in Figure 1 to present the results.
Answer: In order to respect the results order, we have redrawn Figure 1. Changes are made also in the description of the different tests on the material and methods. Please see changes made in the revised version.
- Lines 183 197: the authors did not evaluate the effects, but the differences between before and after Ramadan.
Answer: Correction made as suggested, Please see changes made in the revised version.
- Moreover, they did not investigate fasting (such as hours since last meal) or use the fasting condition as a covariate, and it could not maybe be correct saying "effects of Ramadan fasting."
Answer: “Ramadan fasting” was changed by “Ramadan observance”. Please see changes made in the revised version.
- The punctuation should be more consistent: sometimes there is the space between = and sometimes not. Furthermore, P and D should be in italics.
Answer: The t-value, p-value and Cohen´s d were included in the tables for each parameter as suggested by the reviewer 3.
- Tables are not recalled in the text.
Answer: As you recommended, the tables are recalled in the text. We added the following sentences in the revised version:
“The results of the POMS are illustrated in Table 1.”
“In Table 2, we provided the results of the PSQI questionnaire.”
“The obtained results of the subjective responses are provided in the table 3 illustrated below.”
- Line 194: why the authors wrote during and not BR; maybe they should be more precise in defining the evaluated periods.
Answer: The mistake is corrected as you recommended and the sentence is modified as follows:
“Sleep quality scores were higher at ER compared to BR (p = 0.01, d = 0.88).”
Please see changes made in the revised version.
- Table 2: the authors partially used PSQI components and partly the free-answers of the questionnaire. Maybe, it may be coherent to report all the seven components in addiction to sleep latency, duration and efficiency.
Answer: The seven components of the PSQI were now reported in the results section.
Please see changes made in table 2.
- Subjective responses to the questionnaires:
- The title is equivocal because all the answer are subjective;
Answer: The title was changed to be “Perceptual measures”.
- Sleep duration and quality are part of the PSQI questionnaire;
Answer: Sleep duration and quality recorded here represented the night preceding the test session. However, the PSQI questionnaire was used to assess subjective sleep quality over the previous month.
- KSS results are reported without being explained in the methods section.
Answer: KSS is removed and we opt only for the ESS questionnaire since the two questionnaires have the same purpose.
- Are these data referring to the visual analogue scale?
Answer: Yes, these data were referring to the visual analogue scale.
- 5 Decision-making (response accuracy): in this paragraph, also reaction time analysis is included; thus, the title is not inclusive of all the reported data.
Answer: According to this recommendation, title is corrected and it becomes: “Decision-making: response accuracy and reaction time”
- Figure 2:
- it is not recalled in the text;
Answer: The figure is recalled in the revised version as you recommended and the following sentence is added: As it is illustrated in Figure 2, statistical analysis…….(please see section 3.5)
- the measuring unit is missing;
Answer: The figures represent the number of correct responses.
- bars should be of different colours.
Answer: Correction made as suggested. Please see changes made in the revised version.
DISCUSSION
- Line 224, Another time, the authors evaluated the differences between ER and BR rater than the effects.
Answer: Correction made as suggested. Please see changes made in the revised version.
- Line 240, why did the authors used during and not ER?
Answer: The authors used during and not ER because the PSQI questionnaire was used to assess subjective sleep quality over the previous month and not only for the ER.
- Lines 245-248, did the authors write this sentence referring to their data or those of another study. In the case of the first hypothesis, this could not be said based only on the showed statistical analysis.
Answer: We have now added the references for these sentences. Please see changes made in the revised version.
- Line 264, the authors did not evaluate the association in the statistical analysis.
Answer: We have modified the sentence as follows:
“However, in the present study, the decrease of the decision time at the ER in comparison with BR was not accompanied with an improvement of decision making.”
Please see changes made in the discussion.
- Line 271, authors did not conduct an intervention study; thus, they did not need a control group excluding the possibility of a not-Muslim or not-observing Ramadan control group.
Answer: We have removed this sentence from the limitations of the study. Please see changes made in the revised version.
- The authors should consider that their analysis does not involve women, and their data and conclusion only refer to men population.
Answer: We have added this sentence:
“The results of the present study could not be generalized to all populations. In fact, our data and conclusions only refer to the men population.”
Please see changes made in the revised version.
- Also, time of day could influence performance, attention and fatigue. I suggest the authors also consider the following papers: Mulè et al., 2020, doi: 10.1007/s11332-019-00610-9; Roveda et al., 2020 doi: 1080/07420528.2020.1729787; Montaruli et al., 2019, doi: 10.1080/07420528.2019.1652831.
Answer: As suggested by the reviewer, we have added these references in the revised version.
Reviewer 2 Report
Has the test of scenarios with kung-fu fights been validated? Are there any previous studies on this validation? The actual scenario of a fight may allow for more than one suitable action. I believe that authors need to better explain these aspects.
When the authors mention the use of the Student's t test, I think they should mention whether they used paired samples or the independent samples. I think it must be paired samples. In addition, it is necessary to explain whether the statistical calculation performed was for a one-tailed or two-tailed test.
How did the authors consider the magnitude of the effect (d value)? For example: Very small, Small, Medium, Large, etc.
Author Response
Reviewer 2:
- Has the test of scenarios with kung-fu fights been validated? Are there any previous studies on this validation? The actual scenario of a fight may allow for more than one suitable action. I believe that authors need to better explain these aspects.
Answer: The test of scenarios with kung-fu fight was not validated before. However, in the present study, the 35 scenarios were chosen with the collaboration of two Kung-Fu expert coaches.
- When the authors mention the use of the Student's t test, I think they should mention whether they used paired samples or the independent samples. I think it must be paired samples.
Answer: Correction made as suggested. Please see changes made in the statistical analysis.
- In addition, it is necessary to explain whether the statistical calculation performed was for a one-tailed or two-tailed test.
Answer: The statistical calculation was performed for a two-tailed test. Please see changes made in the text.
- How did the authors consider the magnitude of the effect (d value)? For example: Very small, Small, Medium, Large, etc.
Answer: The following sentence was added:
“Cohen’s d of 0.2, 0.5 and 0.8 represent small, moderate, and large effect sizes, respectively (Cohen 1988).”
Also, the interpretation was added in the results part.
Please see changes made in the revised version.
Reviewer 3 Report
Dear authors,
I want to congratulate you for the novelty and interest of this study. Therefore, authors have studied the effect of Ramadan Intermittent Fasting (RIF) on mood, sleep and decisions making task in combat sports trained athletes. Considering the elevated numbers of Muslims athlethes, to know the effect of RIF on sleep and cognitive performance is very useful for athletes and trainers. The study presents a high quality and a good procedure. In addition, the manuscript is well written. Nevertheless, the study presents some aspects that could be improved it. Therefore, I recommend to make the next changes:
- It´s necessary to include the same order between the Figure 1, the description of the different tests on the material and methods and the results. Therefore, ESS need to be the first test described and the first on the results section, then POMS, etc.
- Chi squared test: I propose to perform this test for detected possible different normative values (i.e., poor and good sleep qualities on PSQI test) at the pre and post Ramadan. Therefore, it´s very interesting to analyse possible statistical difference between athletes with different categorization before and after Ramadan.
- Odd ratio (OR) with interval of confidence: in addition to the previous analysis, it could be very interesting to include the OR for detecting a possible relative risk of present a low quality of sleep after the Ramadan.
- I propose to include the t-value, p-value and Cohen´s d on the tables for each parameter. Nevertheless, authors need to consider include additional tables or figures for the analysis of the chi squared test and OR calculation.
- An impairment of sleep could affect to the fatigue and supercompensation response on well trained athletes. Therefore, I suggest to make reference to it on the Discussion section.
Author Response
Reviewer 3:
I want to congratulate you for the novelty and interest of this study. Therefore, authors have studied the effect of Ramadan Intermittent Fasting (RIF) on mood, sleep and decisions making task in combat sports trained athletes. Considering the elevated numbers of Muslims athlethes, to know the effect of RIF on sleep and cognitive performance is very useful for athletes and trainers. The study presents a high quality and a good procedure. In addition, the manuscript is well written. Nevertheless, the study presents some aspects that could be improved it. Therefore, I recommend to make the next changes:
- It´s necessary to include the same order between the Figure 1, the description of the different tests on the material and methods and the results. Therefore, ESS need to be the first test described and the first on the results section, then POMS, etc.
Answer: Correction made as suggested. Please see changes made in the revised version.
- Chi squared test: I propose to perform this test for detected possible different normative values (i.e., poor and good sleep qualities on PSQI test) at the pre and post Ramadan. Therefore, it´s very interesting to analyse possible statistical difference between athletes with different categorization before and after Ramadan.
Answer: The reviewer is right; however, in the present study we aimed only to compare differences between BR and ER. This way we didn’t perform a Chi squared analysis.
- Odd ratio (OR) with interval of confidence: in addition to the previous analysis, it could be very interesting to include the OR for detecting a possible relative risk of present a low quality of sleep after the Ramadan.
Answer: The confidence interval were included in the revised version. Please see changes made in the revised version.
- I propose to include the t-value, p-value and Cohen´s d on the tables for each parameter.
Answer: Correction made as suggested. Please see changes made in the revised version.
- Nevertheless, authors need to consider include additional tables or figures for the analysis of the chi squared test and OR calculation.
Answer: Please see our responses to question 3 and 4.
- An impairment of sleep could affect to the fatigue and supercompensation response on well trained athletes. Therefore, I suggest to make reference to it on the Discussion section.
Answer: This sentence was added in the discussion session. Please see changes made in the revised version.
Reviewer 4 Report
Dear authors,
Thanks for the submission.
You will find attached my detailed review, 43 comments, up to the Discussion section, as I didnt go forward due to concerns that I had up to that point.
Briefly, the title doesnt actually reflect the study and vice versa. Methods are not appropriate to answer your RQ, which included too many stuff that are not listed in the title. With all that you have you can write two proper manuscripts that deliver your message and not trying to squeeze everything in one attempt that currently is failing to deliver the message.
From the title I see 3 pillars that need to be addressed in the Intro, Methods Results and Discussion. This is the Ramadan, Kungfu - combat sport- and decision making. Any other inference to sleep etc is out of the scope of this manuscript titled as "Ramadan observance impairs Kung-Fu-specific decision-making skills "
Intro doesnt connect proper these 3 pillars to lead the reader to the purpose and the need for this study. Methods have failed to capture - explain properly the scientific soundness of the research design to address the RQ. The issue that I see here is that these data were collected in 2019, probably for a different purpose, and now a re-examination of these already collected data were evaluated to 'see" if it is possible to "publish" something extra. Doing as such has many implications on the writing process and of course on the scientific soundness of the "applied" research design in order to answer another question using a methodology not designed for that purpose.
Is this a common practice? Yes it is , we see it constantly, but it is up to the authors to present a RQ that can be answered with a methodology that was designed to answer another question.
I noticed also that citations were used that did not actually examined what it was stated that they did. Also, the GPower info is lacking.
Due to the inability to present a RQ that can be answered with this research design I am suggesting a rejection and I advise to rewrite this from the scratch and submit it again as two papers that actual can pass your message and drive the field vertically.

Author Response
Reviewer 4:
Thanks for the submission.
You will find attached my detailed review, 43 comments, up to the Discussion section, as I didnt go forward due to concerns that I had up to that point.
Briefly, the title doesnt actually reflect the study and vice versa.
Answer: The title was changed. Please see changes made in the title.
Methods are not appropriate to answer your RQ, which included too many stuff that are not listed in the title. With all that you have you can write two proper manuscripts that deliver your message and not trying to squeeze everything in one attempt that currently is failing to deliver the message.
Answer: The title was changed. Also, some explanations were added in the methods section. Please see changes made in the text.
From the title I see 3 pillars that need to be addressed in the Intro, Methods Results and Discussion. This is the Ramadan, Kungfu - combat sport- and decision making. Any other inference to sleep etc is out of the scope of this manuscript titled as "Ramadan observance impairs Kung-Fu-specific decision-making skills "
Answer: The reviewer is right; however, the other parameters were investigated to establish possible relationships and explanation for the difference between BR and ER for the decision making.
Intro doesnt connect proper these 3 pillars to lead the reader to the purpose and the need for this study. Methods have failed to capture - explain properly the scientific soundness of the research design to address the RQ. The issue that I see here is that these data were collected in 2019, probably for a different purpose, and now a re-examination of these already collected data were evaluated to 'see" if it is possible to "publish" something extra. Doing as such has many implications on the writing process and of course on the scientific soundness of the "applied" research design in order to answer another question using a methodology not designed for that purpose. Is this a common practice? Yes it is , we see it constantly, but it is up to the authors to present a RQ that can be answered with a methodology that was designed to answer another question. I noticed also that citations were used that did not actually examined what it was stated that they did. Also, the GPower info is lacking.
Answer: This study was conducted to examine principally the difference between BR and ER for decision making (the first study in this field) and no re-examination of the data was realised. This study was conducted in Ramadan 2019 and the remaining period until submission was the period of data analysis, written the first draft and proofreading of the manuscript by all authors until the validation of the submitted version.
Due to the inability to present a RQ that can be answered with this research design I am suggesting a rejection and I advise to rewrite this from the scratch and submit it again as two papers that actual can pass your message and drive the field vertically.
Answer: We perform some changes in all parts of the paper. Hope this version is appropriate for publication.
Round 2
Reviewer 1 Report
The manuscript significantly improved after the revision.
I recommend only two more corrections:
- In my opinion, in the manuscript should be better specify that PSQI evaluates sleep during Ramadan while perceptual measures evaluate sleep at the end of Ramadan;
- Measure unit is missing in Figure 2.
Author Response
Reviewer 1
The manuscript significantly improved after the revision.
I recommend only two more corrections:
- In my opinion, in the manuscript should be better specify that PSQI evaluates sleep during Ramadan while perceptual measures evaluate sleep at the end of Ramadan;
Answer: This information was inserted in the revised version. Please see changes made in the text.
- Measure unit is missing in Figure 2.
Answer: This information was inserted in the revised version. Please see changes made in the text.
Reviewer 3 Report
Dear authors,
Thank you very much for the effort made on the changes proposed. I believe that this good manuscript have improved, however, I propose to consider the next minor comments:
- POMS is abbreviated twice (lines 59 and 128) and must be abbreviated only the first time (line 59).
- I recommend to perform the chi squared test for analysing possible statistical differences between athletes with different categorization before and after Ramadan.
- The odd ratio has not been analysed. Therefore it could be an additional analysis to the chi squared test.
Author Response
Reviewer 3
Dear authors,
Thank you very much for the effort made on the changes proposed. I believe that this good manuscript have improved, however, I propose to consider the next minor comments:
- POMS is abbreviated twice (lines 59 and 128) and must be abbreviated only the first time (line 59).
Answer: This information was inserted in the revised version. Please see changes made in the text.
- I recommend to perform the chi squared test for analysing possible statistical differences between athletes with different categorization before and after Ramadan.
Answer: We appreciate the reviewers’ suggestion here. However, in our study we employed the Wilcoxon signed-rank test as we sought to compare repeated measurements on a single sample to discern whether their population mean ranks differ. Indeed, we see the merit in the reviewers’ approach, however, we were not powered to further segment our sample into categories; therefore, it would not be appropriate, in this instance to investigate categorized distributions amongst our data.
- The odd ratio has not been analysed. Therefore it could be an additional analysis to the chi squared test.
Answer: With regard to this comment, related to the above response, we did not power this study to be able to further categorize our sample; therefore additional odds ratio analysis would be inappropriate, misleading, or spurious. However, we absolutely acknowledge the importance of being able discern OR in this regard, and have suggested that additional work, suitably powered for such analyses, be conducted,
Reviewer 4 Report
Dear Authors,
Thanks for the revised versions. Even though the topic is interesting, still the way it is presented doesnt warrant acceptance.
I have highlighted that you cannot in 5000 word limit to cover adequately a paper that covers 1-IV and 5 DVs! Pick one and stick to this and also use the proper tools/instruments to assess changes in your DV. You need also in the Intro to connect the topics for the Reader, to educate about the problem and what is the thing that this study bringing in the Literature.
Methods are chaotic as no clear explanation is given why a multidimesional tool was used to assess one specific dimension - like POMS to assess Fatigue? Why multiple tools were used to assess the same element, for example Total Sleep time with PSQI and ESS? The research design is not adequate enough to answer the RQ - Ramadan observance differences in decision making in KungF - as the sample size cannot be justified using the Gpower, there is no homogeneity in the sample, no control for sleeping time, eating times, sleeping habits, training times, training volumes, etc a myriad of things that may have influenced their responses is missing or not reported.
Moreover, the study was conducted in the 2019 and the IRB approval was obtained in 2020...
Results do not repeat info but more info is needed in the tables like what kind of tool was used. Also why results are given in T and Z scores? How can the reader reach to a conclusion when the same DV - total sleep time is reported in
7.8±1.1 6.3±1.9* Z = 2.41 0.01 1.26 (large effect) 0.42 – 2.64
7.3±1.2 6.1±1.8* effect) Z = 2.13 0.03 0.78 (large 0.004 – 0.09
Discussion needs work to discuss the findings

Author Response
Reviewer 4
Dear Authors,
Thanks for the revised versions. Even though the topic is interesting, still the way it is presented doesnt warrant acceptance.
I have highlighted that you cannot in 5000 word limit to cover adequately a paper that covers 1-IV and 5 DVs! Pick one and stick to this and also use the proper tools/instruments to assess changes in your DV. You need also in the Intro to connect the topics for the Reader, to educate about the problem and what is the thing that this study bringing in the Literature.
Methods are chaotic as no clear explanation is given why a multidimesional tool was used to assess one specific dimension - like POMS to assess Fatigue? Why multiple tools were used to assess the same element, for example Total Sleep time with PSQI and ESS? The research design is not adequate enough to answer the RQ - Ramadan observance differences in decision making in KungF - as the sample size cannot be justified using the Gpower, there is no homogeneity in the sample, no control for sleeping time, eating times, sleeping habits, training times, training volumes, etc a myriad of things that may have influenced their responses is missing or not reported.
Moreover, the study was conducted in the 2019 and the IRB approval was obtained in 2020...
Results do not repeat info but more info is needed in the tables like what kind of tool was used. Also why results are given in T and Z scores? How can the reader reach to a conclusion when the same DV - total sleep time is reported in
7.8±1.1 6.3±1.9* Z = 2.41 0.01 1.26 (large effect) 0.42 – 2.64
7.3±1.2 6.1±1.8* effect) Z = 2.13 0.03 0.78 (large 0.004 – 0.09
Discussion needs work to discuss the findings
Answer: As suggested by the reviewer, we do our best to improve all parts of the manuscript. We, hope that this version is acceptable by the reviewer. Otherwise, we are ready to do all the necessary changes.
For the questionnaires and scales, the Arabic version was utilized when it was validated. Otherwise, we utilize the French version as the second language in our country is the French. If there is no Arabic or French version we used the English version and in our study the participants understand the English language (third language in our country and is included in from the school studies).
The ethical number is correct. However, the reviewer is right as the date was inserted by mistake. This was corrected in the revised version.